# Rescuing low frequency variants within intra-host viral populations directly from Oxford Nanopore sequencing data

Yunxi Liu[1,3], Joshua Kearney[1,3], Medhat Mahmoud [2], Bryce Kille[1], Fritz J. Sedlazeck [1,2] & Todd J. Treangen [1✉]

Infectious disease monitoring on Oxford Nanopore Technologies (ONT) platforms offers rapid turnaround times and low cost. Tracking low frequency intra-host variants provides important insights with respect to elucidating within-host viral population dynamics and transmission. However, given the higher error rate of ONT, accurate identification of intra-host variants with low allele frequencies remains an open challenge with no viable computational solutions available. In response to this need, we present Variabel, a novel approach and first method designed for rescuing low frequency intra-host variants from ONT data alone. We evaluate Variabel on both synthetic data (SARS-CoV-2) and patient derived datasets (Ebola virus, norovirus, SARS-CoV-2); our results show that Variabel can accurately identify low frequency variants below 0.5 allele frequency, outperforming existing state-of-the-art ONT variant callers for this task. Variabel is open-source and available for download at: www.gitlab.com/treangenlab/variabel.

[1] Department of Computer Science, Rice University, 6100 Main Street, Houston, TX 77005, USA. [2] Human Genome Sequencing Center, Baylor College of Medicine, Houston, TX 77030, USA. [3] These authors contributed equally: Yunxi Liu, Joshua Kearney. ✉email: treangen@rice.edu

Oxford Nanopore Technology (ONT) has become a dominant technology for rapid sequencing of COVID-19 patients due to its low cost and relatively simple preparation methods[1]. ONT datasets have proliferated during the pandemic; there are now well over 100,000 sequenced COVID-19 samples from ONT alone in the NCBI SRA database and over a half a million SARS-CoV-2 genomes assembled from ONT[2]. Intra-host variation of COVID-19 reveals important information about many aspects of the disease, such as future variants of concern and the response to different treatments[3–6]. When SARS-CoV-2 infects a human host, a combination of viral and host proteins facilitates the replication of the virus[7]. Intra-host variants then arise during the expansion of the intra-host viral population and homologous recombination[8], some of which may be biologically relevant[9]. Given the ubiquity of COVID-19 ONT data, the goal of our study is to explore the use of widespread ONT data for detection of intra-host variation to elucidate currently "hidden" biology. However, due to the relatively high error rate of ONT, ranging from 5% to 15%, true variation within hosts is obscured by sequencing errors contained in the raw data[10]. Our assumption is that the allele frequency of true SNV within a sample is subject to change across samples, while those of sequencing errors are independent of the sample and thus are highly stable/similar. This is especially the case for similar basecallers and flow cells versions. Multiple studies have shown that the allele frequency of a true variant would experience a significant change over time or over samples collected from different patients[5,11–13]. Indeed, the size of the population of SARS-CoV-2 virions within a host undergoes exponential growth post infection, increasing from a handful of virions to one billion virions or more[14]. Furthermore, the vast majority of sequencing errors in ONT data are deletions, related to homopolymer regions where the same nucleotide occurs consecutively, or low-complexity regions[15–17].

Current read-based error correction and polishing methods for ONT data primarily target genome assembly and haplotype-based variant detection[18–20]. Raw read polishing has proven to be extremely effective in generating high quality assemblies; however, information supporting low frequency (less than 0.5) intra-host variants is almost always lost during the process. An alternative approach to preserving intra-host variation during error correction involves integrating haplotype information into the assembly step[21,22]. Strainline uses a combination of local De Bruijn graph assembly and overlap extending to generate haplotype genomes[21]. CliqueSNV constructs haplotype sequences by recognizing linked SNVs that are supported by a single read[22]. While both methods assemble genomes at strain level resolution, haplotype phasing from ONT sequencing protocols for SARS-CoV-2 is challenging due to the limited read length from amplicon sequencing (250–500 bp)[23], uneven coverage, and susceptibility to bias from single nucleotide variation. Furthermore, sequencing error in ONT data is context dependent[17,24].

Here, we present Variabel, a novel variant call filtering tool that is able to recover intra-host variants from ONT data alone, for the first time, by exploiting the tendency of true variants to change in allele frequency across samples. The key concept behind Variabel is that by leveraging information from viral population dynamics, we can distinguish the true variants from sequencing errors caused by ONT by comparing samples collected across different time points or samples collected from different patients. Variabel is constructed as a series of filters that operate on the variant call format (VCF) files returned by an existing variant caller. Figure 1 illustrates the variant calling workflow and the algorithms of Variabel. It includes an allele frequency variation filter, which identifies true variants that are shared across different samples (see Fig. 1B) and an insertion/deletion (indel) filter that identifies false indel calls based on Shannon's entropy values of the region near indel sites (see Fig. 1C).

## Results

We evaluated Variabel via two ONT datasets: (1) a time series COVID-19 dataset and (2) a cross-patient COVID-19 dataset[4,25]. Importantly, samples in both datasets are sequenced with both ONT and Illumina platforms. Both datasets for COVID-19-positive samples are generated using ARTIC v3 primers on the wild-type SARS-CoV-2 genomes at the early stage of the pandemic, where the number of consensus level mutations compared to the reference is low. The time series dataset contains samples taken from an immunocompromised COVID-19 patient over the course of 3 months[4], where 18 pairs of Illumina and ONT sequencing runs passed quality control. The cross-patient dataset includes 154 COVID-19-positive samples collected from patients, and 103 pairs of Illumina and ONT sequencing runs passed our quality control. We selected these two experimental datasets for evaluation as: (i) the time series dataset allows us to track individual changes in allele frequencies over time for a specific patient, and (ii) the cross-patient datasets allow us to explore the utility of Variabel on more readily available SARS-CoV-2 datasets. Variant calls on Illumina sequencing runs by LoFreq[26] are used as a benchmark in our calculation of precision, recall, and F-score. In the benchmark, the time series dataset contains 835 substitutions and 116 indels, and the cross-patient dataset has 2786 substitutions and 757 indels. We also ran Clair3 on the same ONT sequencing runs for benchmarking purposes. While Clair3 is not explicitly designed for virus SNV calling, it represents a state-of-the-art ONT variant caller[27].

Illumina sequencing produces highly accurate reads, which are ideal for intra-host variant calling. On the other hand, while variant calling on ONT sequencing data offers faster turnaround time and is not limited to sequencing centers, it is much more challenging due to a higher error rate in both the sequencing and base calling process. Most of the previously reported intra-host variants have allele frequencies of >0.02 and less than 0.15, which is well above the Illumina error rate but exactly within the expected ONT error rate, highlighting the dichotomy of using one or the other for identification of low frequency intra-host variation. Our results highlight that Variabel is able to call variants in ONT data with high precision. Figure 2 indicates the positions and minimum allele frequencies of variants called by LoFreq and Variabel. By comparing the variant calling results before and after applying Variabel on both the time series dataset (Fig. 2A) and the cross-patient dataset (Fig. 2B), we found that Variabel is able to remove the majority of the false positive calls caused by the sequencing errors of ONT data. The number of variants that are exclusively found in ONT data (marked in red) drop dramatically, while most of the true variants, which are found in both Illumina and ONT reads passed the filters.

We also benchmarked Variabel with Clair3. Figure 3A shows a Venn diagram of variant calls from Variabel and Clair3 compared to Illumina variant calls for 18 time series samples. Out of 415 variant calls made by Variabel, 391 (94.22%) of them are considered true positive calls since they are also found in the Illumina data. Clair3 had a lower number (378) of true positive calls compared to Variabel, and Clair3 had 266 false positives while Variabel only had 24.

Importantly, Variabel is able to rescue true variants in the low frequency domain. Figure 3B shows the false positive (FP) rates at different variant allele frequencies and cumulative density of FP variant calls from Variabel and Clair3 for the time series dataset. First, we see that for ultralow frequency variants (less than 0.1 allele frequency), Clair3 has a FP rate of 100% (all variants identified are false positives), while Variabel's FP rate for these ultralow frequency variants is zero, and Variabel maintained low FP rates for variants with allele frequency below 0.2. Next, we observed that Variabel has much lower FP rate on average for variants with allele frequency below 0.75 compared to Clair3. The

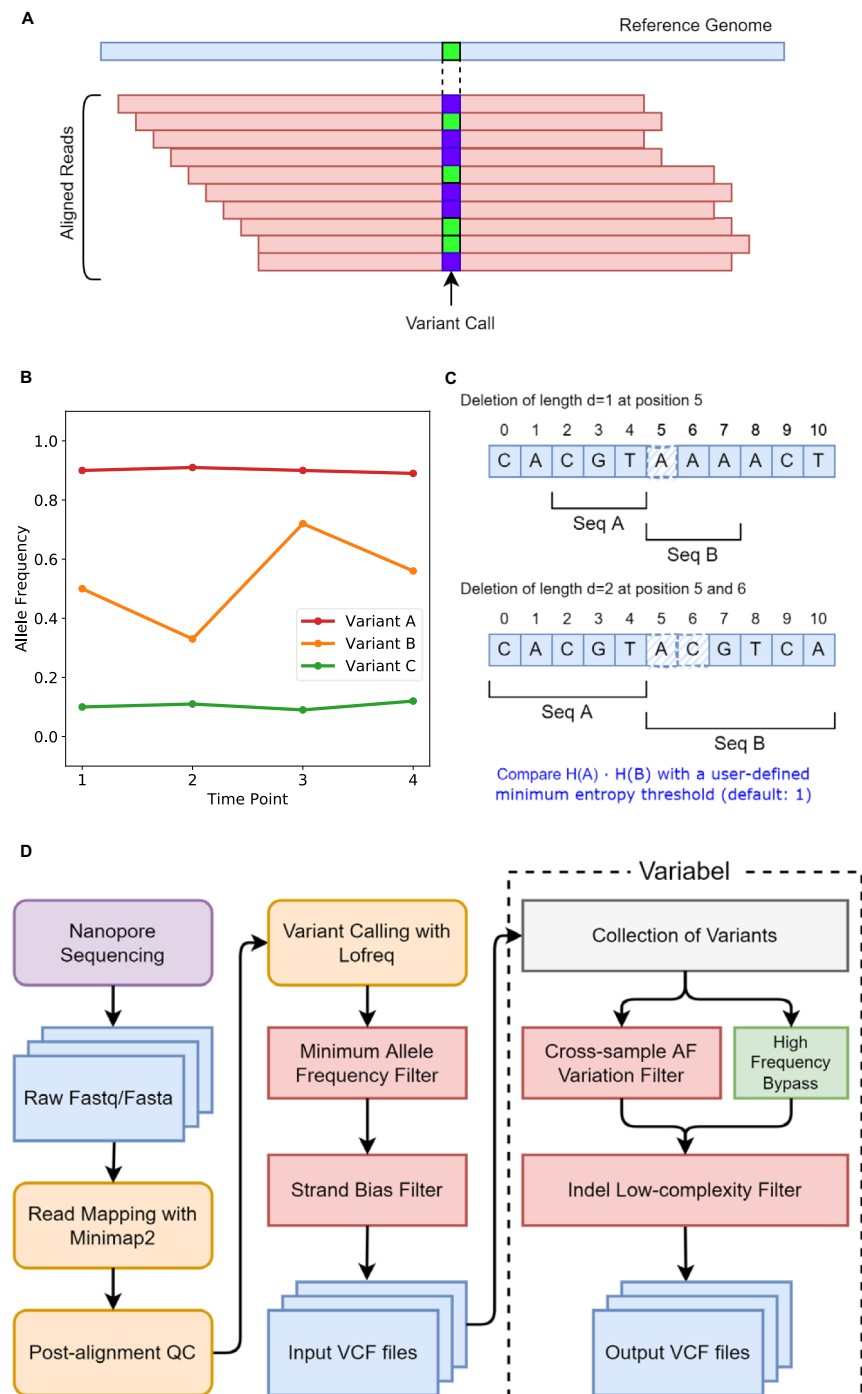

**Fig. 1 Illustration of Variabel algorithm and workflow. A** Sequencing reads from ONT are aligned to the reference genome of SARS-CoV-2 with Minimap2, then variants are called based on the alignments using LoFreq. The figure shows 4 reference supporting (green) reads and 5 alternative supporting reads (blue) amounting to an overall of 55.6% allele frequency within this sample. **B** Cross-sample AF variation filter identifies variants that are shared between samples. Variant calls with maximum AF less 0.65 and maximum AF variation less than 0.05 are classified as false calls. In this example, variant A and B pass the filter while variant C fails. **C** Low-entropy filter calculates the Shannon's entropy H for subsequences (Seq A and B) of the reference genome around the position where the indel call occurs. The length of the subsequences is determined by the length of the indel. Product of two entropy of the subsequences is used to determine whether the indel is a false positive or not. **D** Workflow of Variabel for detecting intra-host variants for ONT sequences.

peak FP rate for Variabel occurs at allele frequency between 0.2 and 0.25, which is associated with Nanopore sequencing errors. Cumulative count plot of FP variant calls shows that Variabel has a close to uniform distribution of false calls along different allele frequencies. On the other hand, more than 70% of the FP calls from Clair3 have allele frequencies less than 0.5.

Figure 3C shows precision, recall, and F-score for variant calls generated by different methods on both the time series and the cross-patient datasets; shown are the LoFreq default output with 0.02 minimum allele frequency filter, Variabel, and Clair3. For all 18 samples that passed quality control from the time series dataset, applying Variabel resulted in a significant mean precision

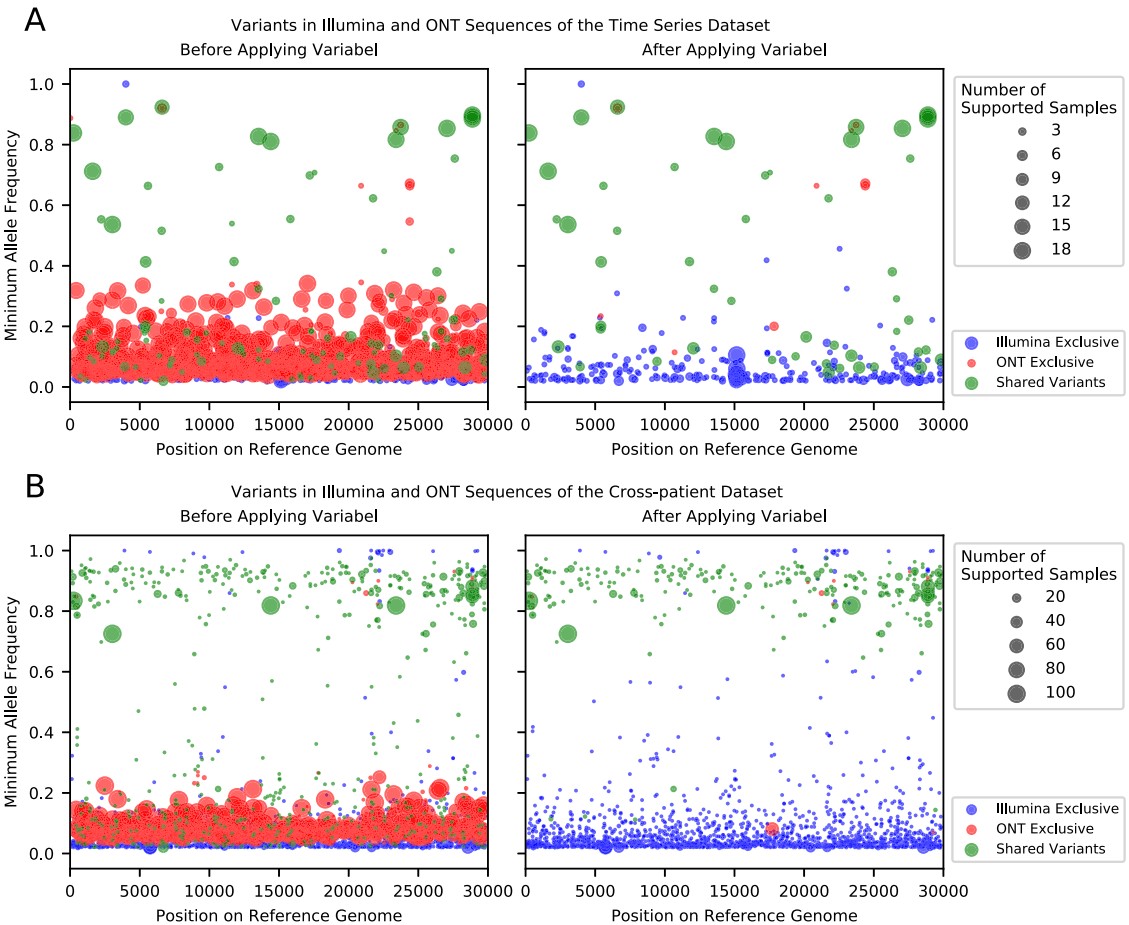

**Fig. 2 Variants called by LoFreq with Illumina and ONT sequences before and after applying Variabel for COVID-19 datasets.** In each of the subfigures, the left plot shows the variant calls before applying variabel and the right plot shows the variant calls after applying variabel. The x-axis shows the position of the variant on the reference genome. The y-axis shows the minimum allele frequency of the variant found in multiple samples. Variants found in the Illumina sequences only are marked in blue, and variants found in the ONT sequences only are marked in red. Variants that are shared between both Illumina and ONT data are shown in green. The size of the dot represents the number of samples supporting the variant. **A** Variant calls of the time series dataset. **B** Variant calls of the cross-patient dataset. For both **A** and **B**, source data are provided as a Source Data file.

increase from 0.042 to 0.940, and a mean F-score increase from 0.076 to 0.599. When applied to the same data, Clair3 had a mean precision of 0.579 and a F-score of 0.489, noticeably under-performing Variabel. Both Variabel and Clair3 had similar mean recall (0.450 for Variabel and 0.440 for Clair3). For all 103 samples that passed quality control (see Methods) from the cross-patient dataset, the mean precision increased from 0.047 to 0.937 after applying Variabel, which is significantly higher than the Clair3 mean precision of 0.897. The mean F-score is 0.640 for Variabel and 0.660 for Clair3. Figure 4A shows the Venn diagram of variant calls for 103 cross-patient samples: Variabel had 69 false positive calls on this dataset while Clair3 had 141. We also tested Variabel with the minimum coverage of the variants set to 10× and 50× (see Supplementary Fig. 1). The precision of the method decreases as the minimum coverage decreases in the cross-patient dataset, since variant calls and their allele frequency estimations at low coverage are less reliable.

We further showed that Variabel can also be used for intra-host variant detection via two non-COVID-19 datasets: (1) an ONT datasets of Ebola virus-positive patient samples[28], and (2) an ONT datasets of norovirus positive patient samples[29]. Figure 5A shows the position and minimum allele frequencies of variant calls before and after applying Variabel on the default LoFreq output with minimum allele frequency set to 0.02 on the Ebola virus dataset. The observation suggested that Variabel is

capable of removing noise while still identifying a large number of low frequency intra-host variants in the samples. Figure 5B shows the same information on the norovirus dataset. The result indicates a high number of consensus level mutations with a few low frequency variants. The filtering rate of Variabel at different allele frequency range is shown in Supplementary Fig. 2.

A synthetic dataset was used as a control to evaluate Variabel's false positive rate[30]. The synthetic should not contain any true biological variants, therefore, all variant calls made should be classified as technical artifacts or errors. Figure 6A indicates that Variabel has a low false positive rate (<0.05) even without a minimum coverage depth threshold, and the false positive rate quickly drops to zero as the minimum coverage depth threshold increases to 200×. We also performed analysis on individual filters of Variabel. Figure 6B shows that the majority of the false variant calls are related to homopolymer errors and can be identified and removed with the low-entropy filter, and the rest of noise is further reduced by the AF variation filter in the synthetic dataset.

## Discussion
Our results highlight that it is possible to accurately identify emerging intra-host variations using ONT sequencing alone. This enables a fast and accurate variant prevalence utilizing the

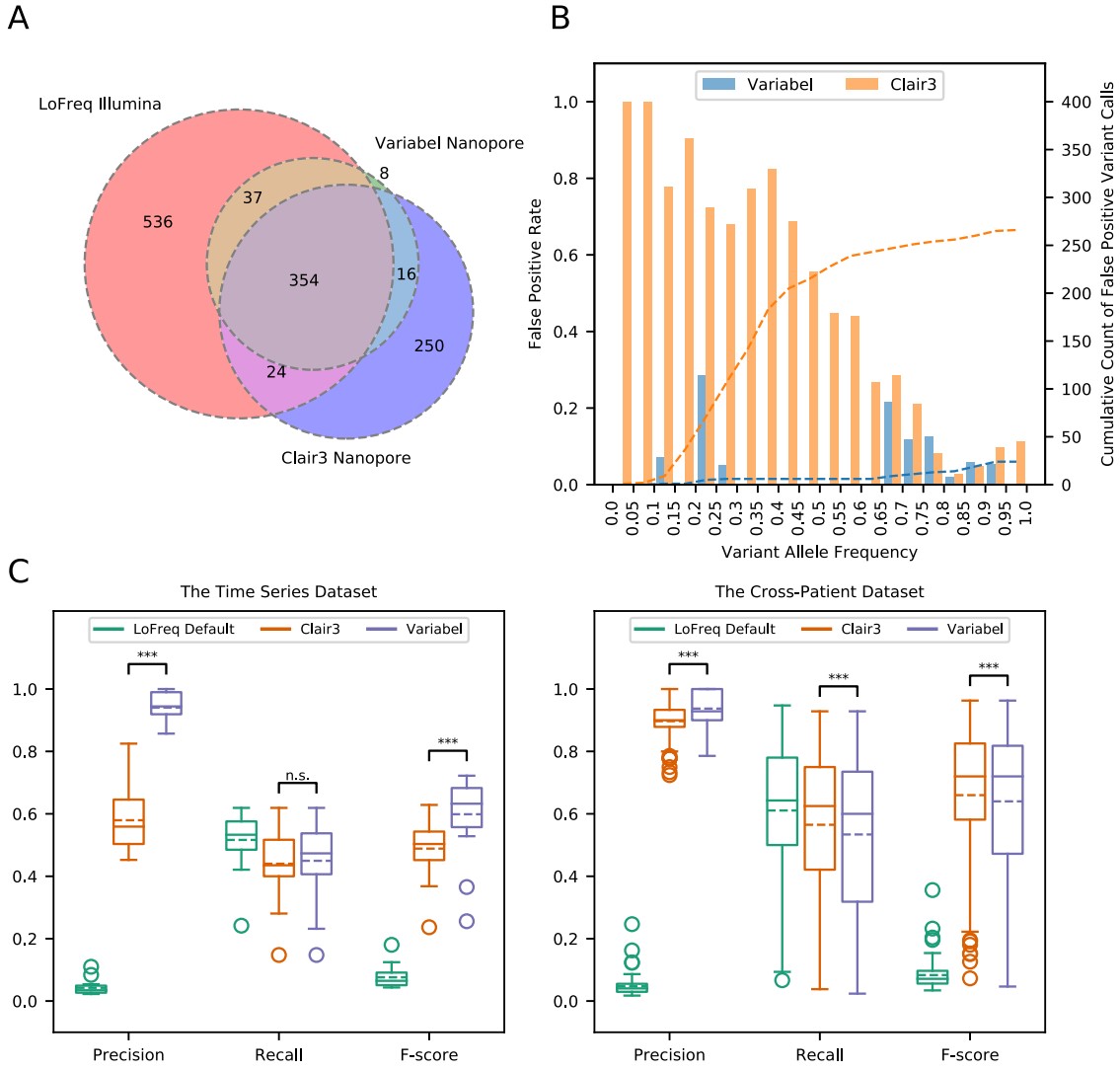

**Fig. 3 Intra-host variant detection on COVID-19 datasets. A** Venn diagram showing counts of variant calls shared between LoFreq on Illumina sequencing runs and Variabel and Clair3 on nanopore sequencing runs on the same samples from the time series dataset. **B** False positive rates at different variant allele frequencies and cumulative count of false positive variant calls of Variabel and Clair3 for the time series dataset. **C** Precision, recall, and F-score comparison of LoFreq default, Clair3, and Variabel on both the time series dataset ($n = 18$ samples from the same COVID-19-positive patient collected over distinct time points) and the cross-patient dataset ($n = 103$ biologically independent samples collected from COVID-19-positive patients). Each box plot includes both median line (solid) and mean line (dashed), and the box bounds the interquartile range (IQR). The Tukey-style whiskers extend from the box by at most 1.5 × IQR. The circle denotes outliers that extend beyond the whiskers. Significance between Clair3 and Variabel were calculated using the two-sided paired *t*-test. Significance labeling: n.s.($P > 0.05$), *($P \leq 0.05$), **($P \leq 0.01$), ***($P \leq 0.001$). The exact *p*-values of the two-sided paired *t*-test of precision, recall, and F-score between Clair3, and Variabel for the time series dataset are $3.36 \times 10^{-11}$, 0.479, and $9.26 \times 10^{-7}$. The exact *p*-values of the two-sided paired *t*-test of precision, recall, and F-score between Clair3, and Variabel for the cross-patient dataset are $1.63 \times 10^{-8}$, $3.86 \times 10^{-9}$, and $4.98 \times 10^{-4}$. For **A**, **B,** and **C**, source data are provided as a Source Data file.

scalability and turnaround time of ONT that is already in place around the world. Variabel uses the variant frequency information and entropy filtering to distinguish true intra-host variants from ONT sequencing error. This is well maintained in the time series data. Furthermore, our experiments have shown that the usage of Variabel can be extended to cross-patient datasets, as well as other non-COVID-19 data, which strongly hints at broader applicability of our approach to the vast amount of ONT based studies.

One of the main limitations of Variabel for cross-patient studies is that the same variant must be observed in at least two samples to activate allele frequency variation filtering. As expected, we observed a drop in average precision in the cross-patient dataset compared to the time series dataset, since samples

collected from different patients are less likely to contain shared variants. Rescuing low frequency intra-host variants is far more challenging for cross-patient data compared to longitudinal data, and the distribution of allele frequencies of true positive variants found by Variabel in the cross-patient dataset clustered above 0.6 while allele frequencies of true positive variants spans in much wider range in the time series dataset (see Fig. 2). Based on a simple simulation (see Fig. 4B) we calculate that ~10,000 samples would be required to recover most of the intra-host variants if we assume variants occur randomly along the genome of SARS-CoV-2. Similarly, we also expect a small drop in performance of Variabel if the time series data included fewer samples (e.g., 2–5). Both scenarios could be improved by leveraging a centralized data depository of low frequency SNV for SARS-Cov-2. Follow-up

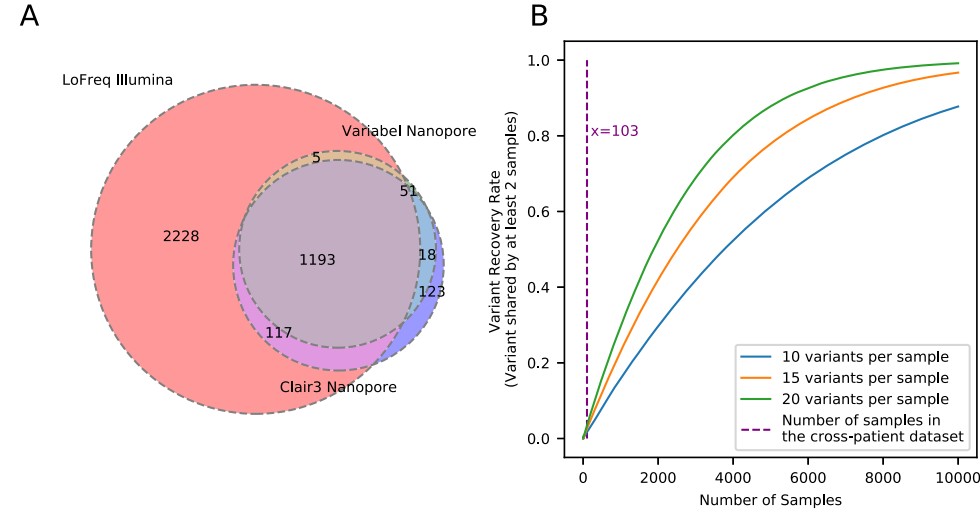

**Fig. 4 Intra-host variant detection on the COVID-19 cross-patient dataset. A** Venn diagram showing counts of variant calls shared between LoFreq on Illumina sequencing runs and Variabel and Clair3 on nanopore sequencing runs on the same samples from the cross-patient dataset. **B** Simulation of fraction of shared variants recovered from different sizes of collections of COVID-19 samples. For both **A** and **B**, source data are provided as a Source Data file.

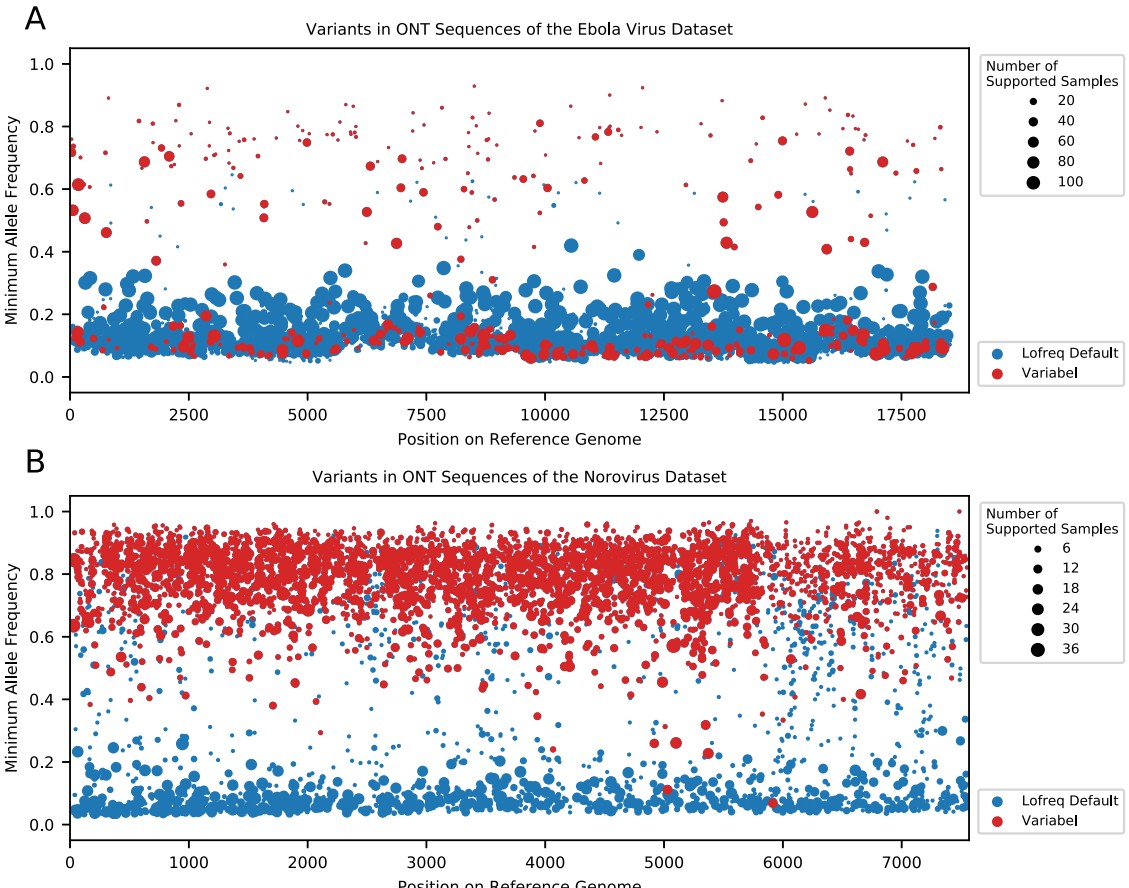

**Fig. 5 Intra-host variant detection on Ebola virus and norovirus datasets. A** Variant calls before and after filtering by Variabel for the Ebola virus dataset. The *x*-axis shows the positions of the variant calls on the reference genome, and the *y*-axis shows the minimum allele frequency the same variant calls among the samples. Variant calls before the filtering are marked in blue and the variant calls after applying Variabel are marked in red. The size of the dot shows the number of samples in which the variant is detected. **B** Variant calls before and after filtering by Variabel for the norovirus dataset. The *x*-axis shows the positions of the variant calls on the reference genome, and the *y*-axis shows the minimum allele frequency the same variant calls among the samples. Variant calls before the filtering are marked in blue and the variant calls after applying Variabel are marked in red. The size of the dot shows the number of samples in which the variant is detected. For both **A** and **B**, source data are provided as a Source Data file.

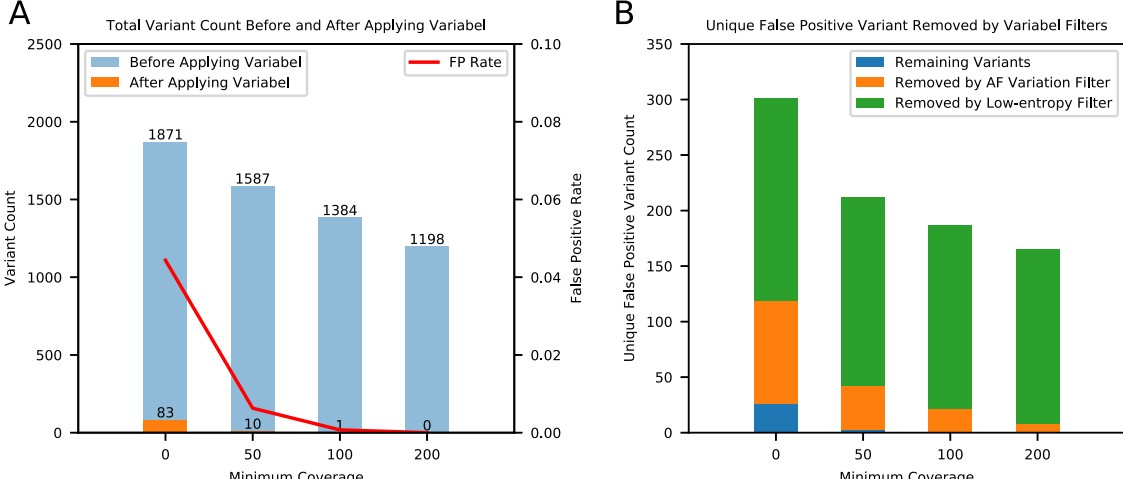

**Fig. 6 False positive rate analysis with the synthetic datasets. A** The bar plot with *x*-axis on the left shows the total number of variant calls in the synthetic dataset before and after applying Variabel. The line with *x*-axis on the right shows the false positive rate at each minimum coverage setting. The variant calls are pre-filtered with 4 different minimum coverage settings (*y*-axis) before applying Variabel. **B** The stacked bar plot showing the number of unique variants removed by different filters of Variabel. The variants removed by the low-entropy filter are shown in green, and the variants removed by the AF variation filter are shown in orange. The remaining false positive calls are shown in blue. For both **A** and **B**, source data are provided as a Source Data file.

studies can then leverage this resource to assess and evaluate the biological importance of observed low frequency variants within and across hosts over time. Given established viral genomic databases such as GISAID are limited only to consensus level sequences[31,32], coordinated community efforts to store and track low frequency variation across vast collections of SARS-CoV-2 datasets will significantly boost Variabel's ability to detect low-AF variants in cross-patient samples.

The intra-host virus mutation rate also affects the performance of Variabel. The virus with high mutation rate during human infection may result in the observation of a higher number of intra-host variants, reducing the number of samples required for AF variant filtering. A recent study on Ebola virus human transmission reported that coincident emergence of a series of variations happening at same locations from different patients has been observed[33], which may explain the large number of low frequency variants we have identified in the Ebola virus dataset (see Fig. 5A).

In this study, we have shown that by comparing the allele frequency of the same variant across multiple samples, Variabel is able to distinguish true biological mutation from ONT sequencing errors, even at low frequency. We believed that there is more potential of the tool outside of the viral domain, such as cancer genomics. The principal would still hold that stable allele frequencies represent artifacts rather than true mutations. As the ONT sequencing gains its popularity over the recent years, we expect to see more highly dynamic ONT sequencing datasets of cross-patient samples or longitudinal samples on a single patient become available in the near future.

Accurate intra-host low frequency variant detection is one of the keys to solving the viral quasispecies genome phasing problem. One common approach to this problem is through de novo assembly[34]. Concordance and discordance of the allele frequencies of variants across multiple samples provides additional information, and can be used to determine whether or not a set of variants is originated from the same viral strain within the population. Therefore, we plan to explore the quasispecies inference problem as one of the future improvements for Variabel.

In conclusion, Variabel is the first method explicitly designed to identify low frequency intra-host variants directly from ONT

data in viral populations. This represents both an important advance for the field and will facilitate the tracking of intra-host variation in COVID-19-positive patients.

## Methods
All data used in this study are downloaded from public databases.

**Dataset descriptions**. Two COVID-19 datasets were used to validate the performance of Variabel. The time series dataset is a longitudinal dataset containing respiratory samples collected from one COVID-19-positive patient with immunodeficiency at 23 different time points across a 101-day period[4]. All samples were sequenced with MinION (Oxford Nanopore Technologies). Among those samples, 20 samples were deep sequenced using the Illumina platform. The ONT data downloaded from NCBI SRA database showed that the quality scores of the reads are corrupted. All bases were assigned with the same quality score "?". The cross-patient dataset contains 154 COVID-19-positive samples that are collected from different patients and sequenced using both Illumina and nanopore platforms[25]. All raw sequencing runs from both datasets are publically available on NCBI SRA database.

We also included two non-COVID-19 ONT sequencing datasets. The Ebola virus dataset is a cross-patient dataset that contains 158 amplicon ONT sequencing runs for Ebola virus-positive patients. The norovirus dataset includes 39 full-length amplicon sequenced cross-patient norovirus GII-positive samples on ONT sequencing platform. All raw sequencing runs from the non-COVID-19 datasets are publically available on NCBI SRA database.

In addition to sequenced patient samples, we used a synthetic dataset to test the false positive rate of Variabel. The synthetic dataset uses control data of a virus-negative nasopharyngeal swab spiked with plasmids containing synthetic S and N genes of SARS-CoV-2 reference genome (NCBI Reference Sequence: NC_045512.2) at concentrations of 0, 10, 100, 500, 1000, and 3000 copies per reaction, with four replicates at each concentration. Each replicate is sequenced using Oxford Nanopore MinION platform for 10, 30 min, 1, 2, and 4 h. The total number of synthetic sequencing runs in this dataset is 112.

**Quality control and read alignment**. We performed pre-alignment quality control on all Illumina sequencing runs using fastp (v0.20.1)[35] with the following command. fastp --detect_adapter_for_pe --cut_front, --cut_window_size 4 --cut_-mean_quality 15 --length_required 15 --qualified_quality_phred 15 --unqualified_percent_limit 40 --n_base_limit 5 --low_complexity_filter. Read alignment for Illumina sequences was done using bwa mem (v0.7.17-r1188) paired end mode with default parameters[36]. Read alignment for nanopore sequences was done using minimap2 (2.20-r1061) with preset map-ont[37]. Alignment files are sorted using samtools (v1.11)[38].

For the time series dataset and the cross-patient dataset of COVID-19 samples, the reference genome used during the alignment is NCBI Reference Sequence: NC_045512.2. For the norovirus dataset, the reads were aligned to the norovirus GII reference genome (NCBI Reference Sequence: NC_039477.1). For the Ebola

virus dataset, the reads were aligned to a reference strain from early in the associated outbreak (GenBank: KR817198.1). For the synthetic data, the reads were aligned to exclusively S and N gene regions of the reference genome (NCBI Reference Sequence: NC_045512.2).

We also performed post-alignment quality control by calculating breadth and depth of genome coverages with samtools depth. For each pair of Illumina and ONT sequencing runs which were generated from the same COVID-19 sample, both sequencing runs must have breadth of genome coverage no less than 0.9 and mean depth of coverage no less than 500, otherwise both sequencing runs are excluded from our experiments. For the norovirus dataset, ONT samples with breadth of genome coverage less than 0.9 and mean depth of coverage less than 500 are excluded from the study, and 37 sequencing runs passed the quality control. For the Ebola virus dataset, since the overall sequencing depth of the samples are low, we lower the minimum mean depth of coverage to 100× but still keep the minimum breadth of genome coverage at 0.9, and 118 sequencing runs passed the quality control. No post-alignment quality control was performed on the synthetic dataset since the dataset is used for coverage depth analysis.

**Variant calling**. We used LoFreq (v2.1.4) to call variants for Illumina samples. This is done by first inserting indel quality score into the BAM files using command "lofreq indelqual --dindel", and then call variants including insertions or deletions (indels) with command "lofreq call --no-default-filter --call-indels". At last, we applied the strain bias filter and removed any variants with allele frequency below 2% or with coverage less than 100× with command "lofreq filter --cov-min 100 --af-min 0.02 --sb-alpha 0.01 --sb-incl-indels"[26]. Both Variabel and Clair3 were used to call variants from nanopore data. We used Clair version 3 [https://github.com/HKU-BAL/Clair3][27] to identify SNVs and indels in samples using default parameters. We used the training dataset specified for ONT and set the --chunk_size to 29,903.

To call variants with variabel, we first stripped the quality score from the nanopore data in order to force LoFreq run with its EM algorithm. Then we insert the indel quality score into the BAM files using command "lofreq indelqual --uniform 16". For the COVID-19 datasets, we used the same command as processing Illumina data to call variants and to filter variants with great strain bias or with allele frequency below 2%. For the Ebola virus and norovirus datasets, a similar process was used except the minimum coverage for filtering variants is set to 10. The collection of VCF files is used as input for Variabel. First, Variabel performs the cross-sample allele frequency variation filtering. It examines each one of the VCF files, identifies variants that are shared between samples, and records their allele frequencies. Any variant with maximum allele frequency less than 0.65 and maximum variation of 0.05 or less across all the samples in which the variant existed is classified as false calls and is eliminated. Variabel then applies a low-entropy filter to any indels that occur in regions of low-complexity. This is designed to eliminate nanopore homopolymer errors that primarily occur in areas with short repeats. Assume a deletion is called at position i on reference genome s with length d, the filter checks the product of the Shannon entropy of the substring s[i-2d: i+1] and the Shannon entropy of the substring s[i+1: i+1+3d]. If the value of the product is less than the user defined threshold (default: 1), the variant is classified as false calls and is eliminated. The variants that pass both cross-sample allele frequency variation filter and low-entropy filter are collected and output in VCF format.

**Primer site masking**. Both the time series dataset and the cross-patient dataset were sequenced with amplicon sequencing method with ARTIC v3 primers (https://github.com/artic-network/primer-schemes/tree/master/nCoV-2019/V3). To reduce the noise of the amplicon sequencing, we masked all variant calls within the primer region with vcftools (v0.1.16) using the command "vcftools --exclude-bed"[39].

**Validation with hybrid COVID-19 datasets**. Since the two COVID-19 datasets we included in our analysis have both Illumina and nanopore sequencing runs in high quality, we used the LoFreq variant calls generated from the Illumina data as a ground truth to evaluate precision, recall, and F-score of Variabel and Clair3.

**Intra-host variant detection in non-COVID-19 datasets**. For the two non-COVID-19 datasets, we compared the variant calls generated by LoFreq before and after applying Variabel to test whether Variabel can be applied on non-COVID-19 datasets. We also calculated the filtering rate of Variabel at different allele frequency range.

**Validation with synthetic dataset**. As the synthetic dataset should not have any true biological variants, all of the variants output by LoFreq are identified as negative. Four different sets of negative variant calls are generated with minimum coverage of the variant calls set to 0×, 50×, 100×, and 200×. We then applied Variabel, the remaining variant calls are identified as false positives, and the ones that have been removed by Variabel are identified as true negatives. We computed the false positive rate of Variabel under those different minimum coverage settings. We also counted the number of unique variants removed by each of the Variabel filters.

**Reporting summary**. Further information on research design is available in the Nature Research Reporting Summary linked to this article.

## Data availability

The source data generated in this study, with the manifest of the datasets, have been deposited in the OSF database with https://doi.org/10.17605/OSF.IO/QBZGP[40]. All sequencing data supporting the findings of this study is publicly available. The time series COVID-19 dataset: NCBI SRA database under BioProject PRJNA682013. The cross-patient COVID-19 dataset: NCBI SRA database under BioProject PRJEB41737. The Ebola virus dataset: NCBI SRA database under BioProject PRJEB10571. The norovirus dataset: NCBI SRA database under BioProject PRJNA713985. The synthetic COVID-19 dataset: China National Center for Bioinformation GSA database with accession number CRA004499. The details about accession numbers of each sequencing run for all datasets used in this study can be found in the manifest. Source data are provided with this paper.

## Code availability

The source code for Variabel is publicly available at: https://gitlab.com/treangenlab/variabel, and we used version 1.0.0 of Variabel for the result and analysis presented in this manuscript[41]. The code used for analysis and figure generation used in this study can be found in: https://doi.org/10.17605/OSF.IO/QBZGP.

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

## Acknowledgements
We would like to thank the contributing authors of the paired Illumina and ONT SARS-CoV-2 sequencing data, which was instrumental for highlighting the benefits of Variabel on ONT data. We would like to thank Dr. Michael Nute for his constructive feedback on the project. M.M. and F.S. were supported by the National Institute of Allergy and Infectious Diseases (Grant#1U19AI144297). T.T. was supported in part by the National Institute of Allergy and Infectious Diseases (Grant#1P01AI152999-01) and by National Science Foundation grant EF-2126387. T.T. and Y.L. were supported in part by C3.ai Digital Transformation Institute COVID-19 award and Centers for Disease Control (CDC) contract 75D30121C11180. This work was supported in part by the Big-Data Private-Cloud Research Cyberinfrastructure MRI-award funded by NSF under grant CNS-1338099 and by Rice University's Center for Research Computing (CRC).

## Author contributions
All authors conceived the experiments, analyzed the results, drafted and revised the paper. Y.L., B.K., J.K., and M.M. conducted the experiments. Y.L. and J.K. wrote the code. F.S. and T.T. managed the study. T.T. supervised the project.

## Competing interests
F.S. received research support from PacBio and ONT. The remaining authors declare no competing interests.
