## [Peer Review File · Nature Communications]

Reviewer comments, first round

Reviewer #1 (Remarks to the Author):

In this paper, the authors present 'Variabel' - a novel approach to calling low-frequency variants from Oxford Nanopore (ONT) sequencing data, which may be used to study intra-host population dynamics in patients infected with SARS-CoV-2 or other viruses. This is a challenging problem due to the relatively high read-level error rate of ONT data. It is an interesting problem because intra-host population dynamics may help us better understand how new strains arise and get transmitted, particularly in immunocompromised patients etc.

Variabel works by filtering candidate variants (SNVs + Indels) identified by an existing tool, Lofreq, across a set of related or unrelated samples. Two main filtering criteria appear to be applied:

- (1) any variant shared between at least 2 samples is excluded if it has a max AF < 0.65 and max variation < 0.05.
- (2) any variant occurring in a low-complexity sequence (defined by Shannon entropy) is excluded.

This approach is quite simplistic, but that's not a bad thing as long as the tool delivers accurate results.

Using Illumina sequencing as a truth set, the authors show that Variabel outperforms Clair (another ONT variant caller) for detecting low frequency variants on two cohorts:

- (1) a time-series cohort of 23 longitudinal samples from a single patient
- (2) a cohort of 145 unrelated samples from different SARS-CoV-2 patients

While Variabel appears to work relatively well on dataset 1 (Fig3a), the outcomes on dataset 2 are not as impressive (Fig4a) and the authors provide modelling that shows thousands of different samples would be required to improve the accuracy of Variabel for cross-patient analysis (Fig4b). This limits the scope of Variabel's use to specific scenarios where many samples from a single patient are available.

The authors propose a solution: by comparing any new sample to a centralised data repository of existing samples, Variabel could work its magic without requiring the user to sequence hundreds/thousands of samples for themselves.

This sounds like a nice solution and I really think the authors should implement it. They suggest that >100,000 sequenced SARS-CoV-2 datasets are available on NCBI SRA. Why not use these to create the centralised database that they have proposed? I realise this is a large undertaking, but it would create a resource that greatly improves the utility of Variabel (assuming the authors' modelling is correct). Without this, the potential use of the tool is too narrow for me to support publication in Nature Communications.

While that is my main suggestion, I have a handful of other ideas that the authors might like to consider, which could potentially improve the potential impact of their paper:

1. The paper only considers SARS-CoV-2 data but should presumably work on other viruses too. Demonstrating this would be worthwhile if the users can find or generate appropriate datasets.
2. I also wonder if Variabel could be used outside viral genomics. Accurate detection and quantitative profiling of low frequency variants could be useful in cancer genomics, metagenomics etc. Showing that Variabel can also be applied in one of these other domains would enhance the paper's impact.
3. Perhaps synthetic controls would be useful for Variabel. Twist Biosciences sells synthetic RNA controls of the SARS-CoV-2 genome. These should contain zero low-frequency variants and inter-sample variation observed results from technical variation / errors. Controls like this might be used to model the error profiles that Variabel takes advantage of for its variant filtering. Variabel

assumes (from what I can gather) that sequencing errors do not vary by more than 5% AF between samples: I think this could be directly measured using synthetic controls. There may also be technical factors (eg number of PCR cycles used) that mean this threshold is different in different experiments, which might be measurable using synthetic controls. I have a limited amount of ONT sequencing data on these controls from Twist that I would be happy to share if the authors are interested, and I'm sure there is plenty more out there.

3. Variabel treats each variant independently, which contrasts haplotype-based methods. I wonder if it would be possible to infer whether different SNPs are within the same, or different, quasi-species based on concordance/discordance of the allele frequencies across samples. SNPs with similar AF patterns across the cohort are likely to be linked, whereas SNPs with different patterns are likely to be unlinked. I'm not sure if this is feasible in practice, but it might be worth thinking about.

Overall, I think Variabel has merits but, without some additional work, the utility is too narrow for a top-quality journal like Nature Communications. I'm sorry I could not be more favourable and I hope that my suggestions prove helpful.

All the best

Dr Ira Deveson (Garvan Institute of Medical Research)

Reviewer #2 (Remarks to the Author):

Liu & Kearney et al present a tool – Variabel – for identifying low frequency variants (<50%) using ONT data. This has been a major challenge in the application of ONT, particularly to studying viral intra-host dynamics, given the higher error rate of ONT compared to Illumina sequencing. Their approach relies on 1) identifying common alleles across multiple samples/timepoints and 2) an error correction algorithm for low-complexity sequences around indels. The performance of Variabel was assessed by comparing ONT & Illumina matched SARS-COV-2 datasets (one longitudinal infection and one cross-patient set). This analysis revealed high precision and the overall strength of this tool, which is in eliminating the majority of variants that are false positives. As reflected on by the authors, the major weakness of this approach is the reliance on identifying shared alleles across samples. This may limit the application of Variabel to specific settings, and that are sampled sufficiently. Despite this, given the use of ONT for viral genome sequencing is only expected to increase, tools such as Variabel should find use and be well-received by the field along with recent improvements in nanopore design that improve performance and error-rates for ONT sequencing.

Some comments for the authors:

1) How does sequence coverage (depth) influence the variant calling and Variabel filtering? Are there any differences in precision/recall etc if the datasets were binned by coverage depth? For example, low (200-500X), moderate 500-1000X, and high (1000X plus) coverage. This might guide some practical thresholds for designing experiments using this tool.

2) The authors haven't shown or discussed how Variabel might be used for viruses others than SARS-CoV-2. As deduced in Figure 4B, the number of samples required in cross-patient studies to identify common alleles can be very high, which could limit its application to these heavily sequenced viruses such as SARS-CoV-2, influenza virus, and HIV. Please reflect on this in the paper and also give consideration to including analysis of datasets from another viral pathogen.

3) The authors should provide the accession numbers for the SRA data used in a table format, and indicate which passed/failed QC. This will enable readers to know exactly which data was used in the analysis.

Reviewer #3 (Remarks to the Author):

Full disclosure, my group generated one of the datasets used to benchmark this software and I'm a consortium author on the other dataset.

The usecase given is SARS-CoV-2 amplicon sequencing (mostly ARTIC) cant reliably call minority variants directly from uncorrected raw nanopore reads. The biggest hurdle being the amplicon coverage varying widely, one could be 20X and the next could be 10,000X combined with a high error rate. This method proposes a population/experiment based strategy to identify minority variants and uses 2 benchmark datasets which have been sequenced with short and long reads. It does address an open challenge.

Because this is a rapidly changing field, for context it should be noted that both datasets were generated with ARTIC V3 on wildtype SARS-CoV-2 genomes, at a time where there was only a small number of mutations in each genome compared to Wuhan Hu-1 (the most commonly used reference).

Inter-host variation has been studied using Illumina with hybrid capture data by Tanya Golubchiks group(<https://pubmed.ncbi.nlm.nih.gov/33688063/>) to great success. Hybrid capture of SARS-CoV-2 is however expensive and only suitable for low Ct samples, so we (the community) do need something which works with amplicon data. I would suggest highlighting this as another selling point of your method.

I have reviewed the source code. It could be streamlined/enhanced by making better use of bcftools (<https://samtools.github.io/bcftools/bcftools.html>), particularly the annotate and filter commands, as it seems you've reinvented some of the steps.

A critical step for variant calling amplicon data is to account for the artificial amplicon sequences which are still present in the ONT reads (they are mostly removed on Illumina through tagmentation but are still be present in low numbers). These are the most well regarded pipelines for processing SARS-CoV-2 ARTIC data:

Illumina: <https://github.com/connor-lab/ncov2019-artic-nf>

ONT: <https://github.com/artic-network/fieldbioinformatics>

You should use the output of the above established pipelines to reanalyse your data as they have been run on over a million samples at this point and every change is poured over in great detail to ensure the results are correct and consistent. They produce VCF files so it should be a straightforward analysis and they are aware of the amplicon sequences. Without this, you will end up seeing artefacts induced by the artificial sequences from the amplicon sequencing. Its a hard problem to tackle with many many edge cases, so you cant do a quick fix on your existing method. I dont think it will fundamentally change the paper but will remove some of the noise, and will lend more confidence to anyone reading the manuscript.

'must be observed in at least two samples to activate allele frequency variation filtering'. This can also be the result of batch level contamination as samples are processed in 96 well plates and multiplexed. An additional feature I think your method brings is that it could be used as a QC check to detect contamination. Amplicon sequencing is extremely prone to contamination.

In Figure 2B 'after applying variable', do the large green dots at the top of the plot fall within the primer regions (https://github.com/artic-network/artic-ncov2019/tree/master/primer_schemes/nCoV-2019/V3)? If so, you should treat them with extreme caution.

Minor comments:

SARS-CoV-2 is the virus, COVID-19 is the disease, so you should correct the terminology. Please note the reference genome you used (Wuhan Hu-1 ?).

LEGEND

Responses in bold **black**

Changes to the manuscript in **blue**

Location of inserted text in (parenthesis)

Added/updated display items indicated in **red**

Reviewer #1

In this paper, the authors present 'Variabel' - a novel approach to calling low-frequency variants from Oxford Nanopore (ONT) sequencing data, which may be used to study intra-host population dynamics in patients infected with SARS-CoV-2 or other viruses. This is a challenging problem due to the relatively high read-level error rate of ONT data. It is an interesting problem because intra-host population dynamics may help us better understand how new strains arise and get transmitted, particularly in immunocompromised patients etc.

R1Q1. This sounds like a nice solution and I really think the authors should implement it. They suggest that >100,000 sequenced SARS-CoV-2 datasets are available on NCBI SRA. Why not use these to create the centralised database that they have proposed? I realise this is a large undertaking, but it would create a resource that greatly improves the utility of Variabel (assuming the author's modelling is correct). Without this, the potential use of the tool is too narrow for me to support publication in Nature Communications.

We thank the reviewer for the positive evaluation of our approach. The reviewer is correct that building a centralized database with intra-host variant information could benefit the broader community, as we described earlier on in our response. We respectfully feel it is well outside the scope of this study, given the computational, database engineering, data harmonization, and data sharing hurdles that would need to all be cleared to enable this effort. Furthermore, we feel this idea has strong merit as a standalone contribution (given no such database exists coupled with the computational, technical, and logistical hurdles).

To dive into this a bit, such an effort would entail running variant calling on over 150,000 samples plus designing, implementing, and testing a database from scratch. While processing such a large number of samples and storing VCF data would be feasible within a couple of months, data harmonization, QC efforts, among other challenges would need to be carefully addressed (such as highlighted by the National COVID Cohort Collaborative: <https://academic.oup.com/jamia/article/28/3/427/5893482> and efforts in Brazil: [https://www.thelancet.com/journals/lanam/article/PIIS2667-193X\(21\)00091-0/fulltext](https://www.thelancet.com/journals/lanam/article/PIIS2667-193X(21)00091-0/fulltext)) To build a high quality, vetted, well architected and open-source database for storing this information, when relying on dedicated time from a database engineer/bioinformatics engineer this can easily take a half of a year alone (and often longer depending upon the complexity of the database and data integration goals). And last but not least, there likely exist data sharing issues tied to SRAs that are linked back

to GISAID samples that would need to be sorted out before building such a resource (to ensure compliance with Ft Lauderdale convention and other data sharing norms such as those raised here: <https://www.nature.com/articles/d41586-021-01194-6>). All combined, we envision building this database would take well over a half year of dedicated effort, likely teetering on a full year of effort to do this well and up to current standards of existing resources that represent stand-alone publications (<https://genomemedicine.biomedcentral.com/articles/10.1186/s13073-020-00822-6>, <https://www.nature.com/articles/s41588-020-0700-8>).

Given these issues, and per the other comments received, in the revised manuscript we focus instead on highlighting the broader applicability of Variabel via the two additional datasets we've included (Ebola virus and Norovirus, two actively monitored viruses), in addition to the rapid growth of ONT data in sequence archives as mentioned by Reviewer #3. The rapid rise in prevalence in ONT data (see figure below, now with nearly 200K ONT datasets), coupled with the focus on infectious disease research and dramatic drop in sequencing costs, will position Variabel to have broad applicability moving forward.

Additionally, based on the observation, we show that Variabel is an innovative method for accurate variant detection with noisy nanopore long reads even on a small dataset with only around hundred samples. Although the power of detecting low frequency (<0.5) intra-host variants is reduced with fewer samples in the dataset, Variabel works well in terms of detecting consensus level mutation, including single nucleotide variants and indels. As outlined below we now also show Variabel's applicability in other viruses.

R1Q2. The paper only considers SARS-CoV-2 data but should presumably work on other viruses too. Demonstrating this would be worthwhile if the users can find or generate appropriate datasets.

We appreciate the reviewer's valuable feedback. The reviewer is correct that Variabel would work on datasets of other viruses. In response to this suggestion, we have performed additional experiments and added two extra datasets. The first dataset contains 142 amplicon sequenced, cross-patient samples for Ebola. We performed quality control on the dataset and removed low quality samples with low breadth and depth coverage that are not suitable for variant calling, and applied Variabel on the remaining 118 samples. The result for this dataset is added to the manuscript as **Figure 5a**. The filtering rate of Variabel at different allele frequencies for the non-COVID datasets is shown in **Supplementary Figure 2**.

Variant calls before and after filtering by Variabel for Norovirus dataset. The x-axis shows the positions of the variant calls on the reference genome, and the y-axis shows the minimum allele frequency the same variant calls among the samples. Variant calls before the filtering are marked in blue and the variant calls after applying Variabel are marked in red. The size of the dot shows the number of samples in which the variant is detected.

Supplementary Figure 2. Total number of variant calls before and after applying Variabel for the Ebola and the norovirus datasets. a) Histogram with x-axis on the left with log10 scale showing total number of variant calls at each allele frequency range before (blue) and after (orange) applying Variabel for the Ebola dataset. The dotted line with x-axis on the right shows the filtering rate of Variabel. **b)** Histogram with x-axis on the left with log10 scale showing total number of variant calls at each allele frequency range before (blue) and after (orange) applying Variabel for the norovirus dataset. The dotted line with x-axis on the right shows the filtering rate of Variabel.

As shown in **Figure 5a**, Variabel works well on the Ebola dataset, detecting not only high frequency consensus level mutation, but also calls numbers of intra-host variants with low allele frequencies for Ebola patients.

The second dataset contains 39 full-length amplicon sequenced cross-patient samples on nanopore sequencing platform. 37 samples passed our quality control step for further investigation. The result for this dataset is added to the manuscript as **Figure 5b**.

Figure 5b shows a much higher number of consensus level mutations with a few intra-host low frequency variants. The high number of the consensus level mutation is mainly because the dataset contains samples that originated from different norovirus strains, and the read alignment was done using the reference genome (NC_039477.1) from NCBI RefSeq to keep the variant calling consistent. As an expected result, the consensus level mutation is dominating in terms of numbers in this dataset.

Both the Ebola dataset and the norovirus dataset shows that Variabel is not limited to COVID samples, but also can be used across other viruses. This broadens the use case of the software and shows that Variabel has much more potential in the future as more nanopore sequencing data become publicly available.

We updated the text in the manuscript to reflect the additional experiments:

> (Main text, Page 5)

We further showed that Variabel can also be used for intra-host variant detection via two non-COVID datasets: 1) an ONT datasets of Ebola positive patient samples, and 2) an ONT datasets of norovirus positive patient samples. **Figure 5A** shows the position and minimum allele frequencies of variant calls before and after applying Variabel on the default lofreq output with minimum allele frequency set to 0.02 on the Ebola dataset. The observation suggested that Variabel is capable of removing noise while still identifying a large number of low frequency intra-host variants in the samples. **Figure 5B** shows the same information on the norovirus dataset. The result indicates a high number of consensus level mutations with a few low frequency variants. The filtering rate of Variabel at different allele frequency range is shown in **Supplementary Figure 2**.

> (Main text, Page 6)

Furthermore, our experiments have shown that the usage of Variabel can be extended to cross patient datasets, as well as other non-COVID data, which strongly hints at broader applicability of our approach to the vast amount of ONT based studies.

> (Main text, Page 6)

The intra-host virus mutation rate also affects the performance of Variabel. The virus with high mutation rate during human infection may result in the observation of a higher number of intra-host variants, reducing the number of samples required for AF variant filtering. A recent study on Ebola human transmission reported that coincident emergence of a series of variations happening at same locations from different patients has been observed, which may explain the large number of low frequency variants we have identified in the Ebola dataset (see Figure 5a).

> (Methods, Dataset Descriptions, Page 13)

Two COVID datasets were used to validate the performance of Variabel.

> (Methods, Dataset Descriptions, Page 13-14)

We also included two non-COVID ONT sequencing datasets. The Ebola dataset is a cross-patient dataset that contains 158 amplicon ONT sequencing runs for Ebola positive patients. The norovirus dataset includes 39 full-length amplicon sequenced cross-patient samples on ONT platform. All raw sequencing runs from the non-COVID datasets are publically available on NCBI SRA database.

> (Method, Quality control and read alignment, Page 14)

For each pair of Illumina and ONT sequencing runs which were generated from the same COVID sample, both sequencing runs must have breadth of genome coverage no less than 0.9 and mean depth of coverage no less than 500, otherwise both sequencing runs are excluded from our experiments. For the norovirus dataset, ONT samples with breadth of genome coverage less than 0.9 and mean depth of coverage less than 500 are excluded from the study, and 37 sequencing runs passed the quality control. For the Ebola dataset, since the overall sequencing depth of the samples are low, we lower the minimum mean depth of coverage to 100X but still keep the minimum breadth of genome coverage at 0.9, and 118 sequencing runs passed the quality control.

> (Methods, Validation with hybrid COVID Datasets, Page 15)

Since the two COVID datasets we included in our analysis have both illumina and nanopore sequencing runs in high quality, we used the lofreq variant calls generated from the illumina data as a ground truth to evaluate precision, recall, and f-score of Variabel and Clair3.

> (Methods, Intra-host Variant detection in non-COVID Datasets, Page 15-16)

For the two non-COVID datasets, we compared the variant calls generated by lofreq before and after applying Variabel to test whether Variabel can be applied on non-COVID datasets. We also calculated the filtering rate of Variabel at different allele frequency range.

R1Q3. I also wonder if Variabel could be used outside viral genomics. Accurate detection and quantitative profiling of low frequency variants could be useful in cancer genomics, metagenomics etc. Showing that Variabel can also be applied in one of these other domains would enhance the paper's impact.

Indeed, we have now incorporated results from non-covid data sets. We did not approach the question if it could be used for cancer research as we deemed it too far outside the scope of our current work. Nevertheless, we believe the reviewer is right given the high dynamic changes across cancer patients or even across time for the same patient. The principal should hold that the fixed AF SNV would represent artifacts rather than true mutations. All of this said, Variabel would still require reasonable coverage to identify and filter low frequency variants of course. We have incorporated several sentences to our manuscript following these suggestions:

> (Main text, Page 6)

In this study, we have shown that by comparing the allele frequency of the same variant across multiple samples, Variabel is able to distinguish true biological mutation from noisy ONT sequencing data, even at low frequency. We believed that there is more potential of the tool outside of the viral domain, such as cancer genomics. The principal would still hold that stable allele frequencies represent artifacts rather than true mutations. As the ONT sequencing gains its popularity over the recent years, we expect to see more highly dynamic ONT sequencing dataset of cross-patient samples or longitudinal samples on a single patient become available in the near future.

R1Q4. Perhaps synthetic controls would be useful for Variabel. Twist biosciences sells synthetic RNA controls of the SARS-CoV-2 genome. These should contain zero low-frequency variants and inter-sample variation observed results from technical variation / errors. Controls like this might be used to model the error profiles that Variabel takes advantage of for its variant filtering. Variabel assumes (from what I can gather) that sequencing errors do not vary by more than 5% AF between samples: I think this could be directly measured using synthetic controls. There may also be technical factors (e.g. number of PCR cycles used) that mean this threshold is different in different experiments, which might be measurable using synthetic controls. I have a limited amount of ONT sequencing data on these controls from Twist that I would be happy to share if the authors are interested, and I'm sure there is plenty more out there.

We very much appreciate the reviewer's offer to provide a synthetic dataset of the SARS-CoV-2 virus. After careful consideration and discussion with the editor, we decided to turn down this offering as we didn't want to cause a potential exclusion of the reviewer. Thus, we identified another synthetic dataset of SARS-CoV-2 virus that is publicly available. The Wang et al 2021 (<https://www.ncbi.nlm.nih.gov/pmc/articles/PMC8420111/>) synthetic dataset contains a set of 112 control samples of a virus-negative nasopharyngeal swab spiked with plasmids containing synthetic S and N genes of COVID-19 at concentrations of 0, 10, 100, 500, 1000, and 3000 copies per reaction, with four replicates at each concentration. Each replicate is sequenced using MinION for 10, 30 min, 1, 2, and 4 h.

As the synthetic sequences should not contain any true biological variation, so all variants observed are caused by artifacts and errors. By using synthetic control data, we were able to identify the false positive calls made by Variabel, as well as exploring the relationship between the false positive rate and minimum coverage of each variant calls.

We added the result in the manuscript as Figure 6a and Figure 6b.

Figure 6. False positive rate analysis with the synthetic datasets. a) The bar plot with x-axis on the left shows the total number of variant calls in the synthetic dataset before and after applying Variabel. The line with x-axis on the right shows the false positive rate at each minimum coverage setting. The variant calls are pre-filtered with 4 different minimum coverage settings (y-axis) before applying Variabel. **b)** The stacked bar plot showing the number of unique variants removed by different filters of Variabel. The variants removed by the low-entropy filter are shown in green, and the variants removed by the AF variation filter are shown in orange. The remaining false positive calls are shown in blue.

Figure 6a shows that Variabel has a reasonably low rate of reporting false positive variant calls. Even without filtering by minimum coverage, we achieved a low false positive rate of 0.044. As the minimum coverage increases, the false positive rate drops dramatically. At minimum coverage of 200x, the false positive rate goes down to zero.

Figure 6b shows the number of unique variants removed by each of the filters in Variabel, and the number of remaining false positive variant calls. A unique variant is defined as a variant with the same position and the allele base among all the samples, regardless of how many times the variant is being observed. This figure shows that most of the false variant calls before applying Variabel in the synthetic dataset are associated with nanopore homopolymer errors and can be identified using our low-entropy filter. Beside that, the majority of the other false calls have been captured by the AF variation filter.

Furthermore, we have updated the following text in the manuscript:

> (Main text, Page 5)

A synthetic dataset was used as a control to evaluate Variabel's false positive rate. The synthetic should not contain any true biological variants, therefore, all variant calls made by should be classified as technical artifacts or errors. **Figure 6A** indicates that Variabel has a low false positive rate (<0.05) even without a minimum coverage depth threshold, and the false positive rate quickly drops to zero as the minimum coverage depth threshold increases to 200x. We also performed analysis on individual filters of Variabel. **Figure 6B** shows that the majority of the false variant calls are related to homopolymer errors and can be identified and removed with the low-entropy filter, and the rest of noise is further reduced by the AF variation filter in the synthetic dataset.

> (Methods, Dataset Descriptions, Page 14)

In addition to sequenced patient samples, we used a synthetic dataset to test the false positive rate of Variabel. The synthetic dataset uses control data of a virus-negative nasopharyngeal swab spiked with plasmids containing synthetic S and N genes of SARS-CoV-2 reference genome (NCBI Reference Sequence: NC_045512.2) at concentrations of 0, 10, 100, 500, 1000, and 3000 copies per reaction, with four replicates at each concentration. Each replicate is sequenced using Oxford Nanopore MinION platform for 10, 30 min, 1, 2, and 4 h. The total number of synthetic sequencing runs in this dataset is 112.

> (Methods, Quality control and read alignment, Page 14)

No post-alignment quality control was performed on the synthetic dataset since the dataset is used for coverage depth analysis.

> (Methods, Validation with Synthetic Dataset, Page 16)

As the synthetic dataset should not have any true biological variants, all of the variants output by lofreq are identified as negative. Four different sets of negative variant calls are generated with minimum coverage of the variant calls set to 0x, 50x, 100x, and 200x. We then applied Variabel, the remaining variant calls are identified as false positives, and the ones that have been removed by Variabel are identified as true negatives. We computed the false positive rate of Variabel under those different minimum coverage settings. We also counted the number of unique variants removed by each of the Variabel filters.

R1Q5. Variabel treat each variant independently, which contrasts haplotype-based methods. I wonder if it would be possible to infer whether different SNPs are within the same, or different, quasi-species based on concordance/discordance of the allele frequencies across samples. SNPs with similar AF patterns across the cohort are likely to be linked, whereas SNPs with different patterns are likely to be unlinked. I'm not sure if this is feasible in practice, but it might be worth thinking about.

Quasi-species genome phasing based on concordance/discordance of the allele frequencies across samples is a very interesting idea and one we would like to explore. While it has potential, we experienced multiple challenges to implement a proof of principle of this concept. Designing such experiments requires carefully crafted synthetic sequencing data with multiple different viral strains of the same species.

Otherwise it would be hard to validate without proper ground truth. Unfortunately, we have not found the suitable dataset to conduct such experiments. We have added the following citations and text to the discussion:

> (Main text, Page 7)

Accurate intra-host low frequency variant detection may be one of the keys to solve viral quasispecies genome phasing problem. One common approach to this problem is through *de novo* assembly³³. Concordance and discordance of the allele frequencies of variants across multiple samples provides additional information, and can be used to determine whether or not a set of variants is originating from the same viral strain within the population. Therefore, we planned to explore the quasispecies inference problem as one of the future improvements for Variabel.

> (References, Page 18)

33. Freire, B., Ladra, S., Paramá, J. R. & Salmela, L. Inference of viral quasispecies with a paired de Bruijn graph. *Bioinformatics* 37, 473–481 (2020).

Reviewer #2 (Remarks to the Author):

Despite this, given the use of ONT for viral genome sequencing is only expected to increase, tools such as Variabel should find use and be well-received by the field along with recent improvements in nanopore design that improve performance and error-rates for ONT sequencing.

We would like to thank the reviewer for the detailed feedback and positive evaluation of our approach.

R2Q1. How does sequence coverage (depth) influence the variant calling and Variabel filtering? Are there any differences in precision/recall etc if the datasets were binned by coverage depth? For example, low (200-500X), moderate 500-1000X, and high (1000X plus) coverage. This might guide some practical thresholds for designing experiments using this tool.

We thank the reviewer for the feedback. The reviewer is correct about allele frequency estimation being less reliable for low coverage regions. Not only the allele frequency estimation but also the variant calls itself become less reliable at low coverage (100x). This is also the reason why samples in each of the datasets have to go through a post-alignment quality control step to remove those with low breadth of genome coverage and those with low mean depth of coverage. To address these concerns, we have performed an extra experiment on a synthetic dataset to test the false positive rate and the relationship between the coverage depth of variant calls and false positive rate. The result is shown in **Figure 6a and 6b. The analysis on the synthetic dataset highlights that Variabel has a low false positive rate even without coverage depth filtering, and as the depth of coverage increase, the false positive rate quickly decreases and reaches zero at**

200x coverage. We believe with 500x mean coverage threshold; the estimation of variant allele frequencies is accurate enough for the AF variation filter to properly work.

With respect to binning samples by coverage, we implemented this suggestion by filtering variants using different minimum coverage. Based on the false positive rate experiment on synthetic data, we have adjusted the minimum coverage for filtering variants from 2x to 100x during our experiment on the COVID datasets to provide more reliable validation, as well as providing analysis on precision, recall, and f-score for the same data with minimum coverage set to 10x and 50x in the Supplementary Figure 1.

For the updated main figures, please refer to R3Q4, since additional filtering on primer sites has also been applied. We also updated the main text and method section to reflect the adjustment.

> (Main text, Page 5)

We also tested Variabel with the minimum coverage of the variants set to 10x and 50x (see Supplementary Figure 1). The precision of the method decreases as the minimum coverage decreases in the cross-patient dataset, since variant calls and their allele frequency estimations at low coverage are less reliable.

> (Methods, Variant calling, Page 15)

At last, we applied the strain bias filter and removed any variants with allele frequency below 2% or with coverage less than 100x with command “lofreq filter --cov-min 100 --af-min 0.02 --sb-alpha 0.01 --sb-incl-indels”.

> (Methods, Variant calling, Page 15)

For the COVID datasets, we used the same command as processing Illumina data to call variants and to filter variants with great strain bias or with allele frequency below 2%. For the Ebola and norovirus datasets, a similar process was used except the minimum coverage for filtering variants is set to 10.

R2Q2. The authors haven't shown or discussed how Variabel might be used for viruses other than SARS-CoV-2. As deduced in Figure 4B, the number of samples required in cross-patient studies to identify common alleles can be very high, which could limit its application to these heavily sequenced viruses such as SARS-CoV-2, influenza virus, and HIV. Please reflect on

this in the paper and also give consideration to including analysis of datasets from another viral pathogen.

We appreciate the reviewer's suggestions. Variabel is not designed exclusively for SARS-CoV-2 datasets and can be applied to other nanopore datasets as well. We added two extra datasets on Ebola virus and norovirus to extend the use case of Variabel to other viral pathogens. The result is shown in the manuscript in **Figure 5a and 5b**.

Depending on the characteristic of the virus itself, the number of samples required to detect the low frequency intra-host variants varies. Study has shown that the mutation rate observed in Ebola virus may rise during human infection, and coincident emergence of a set of SNVs has been observed. This could be the reason why Variabel is still able to detect low frequency SNVs even with a limited number of Ebola samples (shown in **Figure 5a**). On the other hand, our results suggest that even if the number of samples is insufficient to detect low frequency variants, Variabel is still able to make accurate calls on consensus level mutation.

For details on the experiment on non-COVID datasets, please refer to the response to R1Q2.

R2Q3. The authors should provide the accession numbers for the SRA data used in a table format, and indicate which passed/failed QC. This will enable readers to know exactly which data was used in the analysis.

We have updated the manifest for the existing datasets as well as the additional datasets with information on the breadth and mean depth of coverage of each sample, as well as whether or not the sample has passed our quality control and included in this study. The manifest is stored at https://osf.io/qbzgp/?view_only=6cf3b0a15c3e4bfc8529ef725ac660d7.

Reviewer #3 (Remarks to the Author):

Full disclosure, my group generated one of the datasets used to benchmark this software and I'm a consortium author on the other dataset.

The use case given is SARS-CoV-2 amplicon sequencing (mostly ARTIC) can't reliably call minority variants directly from uncorrected raw nanopore reads. The biggest hurdle being the amplicon coverage varying widely, one could be 20X and the next could be 10,000X combined with a high error rate. This method proposes a population/experiment based strategy to identify minority variants and uses 2 benchmark datasets which have been sequenced with short and long reads. It does address an open challenge.

R3Q1. Because this is a rapidly changing field, for context it should be noted that both datasets were generated with ARTIC V3 on wild type SARS-CoV-2 genomes, at a time where there was only a small number of mutations in each genome compared to Wuhan Hu-1 (the most commonly used reference).

We thank the reviewer for the feedback. For an ongoing pandemic, the sample correction period is an important piece of information for variant tracking. We have updated the manuscript to indicate the details on the dataset.

> (Main text, Page 3)

Both datasets for COVID positive samples are generated using ARTIC v3 primers on the wild type SARS-CoV-2 genomes at the early stage of the pandemic, where the number of consensus level mutations compared to the reference is low.

R3Q2. Inter-host variation has been studied using Illumina with hybrid capture data by Tanya Golubchiks group(<https://pubmed.ncbi.nlm.nih.gov/33688063/>) to great success. Hybrid capture of SARS-CoV-2 is however expensive and only suitable for low Ct samples, so we (the community) do need something which works with amplicon data. I would suggest highlighting this as another selling point of your method.

We thank the reviewer for highlighting this and we have now included this citation in the manuscript.

> (Main text, Page 3)

Intra-host variation of COVID-19 reveals important information about many aspects of the disease, such as future variants of concern and the response to different treatments^{3,4,5}.

> (References, Page 18)

5. Lythgoe, K. A. et al. SARS-CoV-2 within-host diversity and transmission. *Science* 372, (2021).

R3Q3. I have reviewed the source code. It could be streamlined/enhanced by making better use of bcftools (<https://samtools.github.io/bcftools/bcftools.html>), particularly the annotate and filter commands, as it seems you've reinvented some of the steps.

We thank the reviewer for their in-depth review of our source code. Our main software deliverable, the Variabel executable, has four functions that perform variant filtering. Three of these functions are unique to the Variabel algorithm and perform filtering across samples, which is a functionality that we do not believe bcftools supports. The fourth function uses LoFreq for basic filtering, and while bcftools may perform the same filtering functionalities as LoFreq, we do not feel that replacing LoFreq with bcftools would make the software more or less streamlined. Similarly with annotation, we already are using existing 3rd party packages. We use SnpEff for annotation in a data preprocessing script that is not part of the Variabel executable.

R3Q4. You should use the output of the above established pipelines to reanalyse your data as they have been run on over a million samples at this point and every change is poured over in great detail to ensure the results are correct and consistent. They produce VCF files so it should be a straightforward analysis and they are aware of the amplicon sequences. Without this, you will end up seeing artifacts induced by the artificial sequences from the amplicon sequencing. It's a hard problem to tackle with many many edge cases, so you can't do a quick fix on your existing method. I don't think it will fundamentally change the paper but will remove some of the noise, and will lend more confidence to anyone reading the manuscript.

Thanks to the reviewer's feedback and the suggestion of using specific pipelines. The focus of this study is to propose a method for post processing of variant calls made by other variant callers. We acknowledge that there might be some edge cases or biases depending on the mixture of pipelines that people use, but the performing benchmark on different variant calling pipelines or different variant callers is out of the scope of this study.

Models for variant calling software have been put under the microscope in recent years. The pipeline and the variant calling tool (Lofreq) we decided to use is designed to balance both accuracy and sensitivity of variant calling. It has a wide range of use cases and is not limited to COVID samples. For details on Variabel results on non-COVID datasets, please refer to the response to R1Q2.

We are aware that amplicon sequencing might introduce noise and artifacts, especially at primer sites. In response to the concern, we updated our method and masked the variant calls in the primer sites for all datasets related to SARS-CoV-2 virus and ARTIC protocol. We updated the **Figure 2a, 2b, 3a, 3b, 3c, 4a** as well as the text in the manuscript.

Thanks to the reviewer's suggestion, after masking the ARTIC v3 primer sites from the previous result, we observed increases of precision and recall for all the benchmarked methods. Variabel achieved greater advantages on the time series data for the COVID patient in terms of precision and f-score, and its precision continued to outperform Clair3 on the cross-patient COVID dataset. There is a noticeable decrease of false positive calls from Clair3 on the cross-patient dataset, resulting in a huge improvement for its f-score.

The following are the updated result figures of the manuscript.

We also have updated the text in the manuscript in concordance with the reviewer’s feedback:

> (Main text, Page 3)

In the benchmark, the time series dataset contains 835 substitutions and 116 indels, and the cross-patient dataset has 2,786 substitutions and 757 indels.

> (Main text, Page 4)

Out of 415 variant calls made by Variabel, 391 (94.22%) of them are considered true positive calls since they are also found in the Illumina data. Clair3 had a lower number (378) of true positive calls compared to Variabel, and Clair3 had 266 false positives while Variabel only had 24.

> (Main text, Page 4)

First, we see that for ultra-low frequency variants (less than 0.1 allele frequency), Clair3 has a FP rate of 100% (all variants identified are false positives), while Variabel's FP rate for these ultra-low frequency variants is around 0.2. Next, we observed that Variabel has much lower FP rate on average for variants with allele frequency below 0.75 compared to Clair3. The peak FP rate for Variabel occurs at allele frequency between 0.2 and 0.25, which is associated with Nanopore sequencing errors. Cumulative count plot of FP variant calls shows that Variabel has a close to uniform distribution of false calls along different allele frequencies. On the other hand, more than 70% of the FP calls from Clair3 have allele frequencies less than 0.5.

> (Main text, Page 5)

For all 18 samples that passed quality control from the time-series dataset, applying Variabel resulted in a significant mean precision increase from 0.042 to 0.940, and a mean f-score increase from 0.076 to 0.599. When applied to the same data, Clair3 had a mean precision of 0.579 and a f-score of 0.489, noticeably underperforming Variabel. Both Variabel and Clair3 had similar mean recall (0.450 for Variabel and 0.440 for Clair3). For all 103 samples that passed quality control (see methods) from the cross-patient dataset, the mean precision increased from 0.047 to 0.937 after applying Variable, which is significantly higher than the Clair3 mean precision of 0.897. The mean f-score is 0.640 for Variabel and 0.660 for Clair3. **Figure 4A** shows the Venn diagram of variant calls for 103 cross patient samples: Variabel had 69 false positive calls on this dataset while Clair3 had 141.

> (Methods, Primer Site Masking, Page 15)

Both the time series dataset and the cross-patient dataset were sequenced with amplicon sequencing method with ARTIC v3 primers (<https://github.com/artic-network/primer-schemes/tree/master/nCoV-2019/V3>). To reduce the noise of the amplicon sequencing, we masked all variant calls within the primer region with vcfTools (v0.1.16) using the command “vcftools --exclude-bed”.

R3Q5. “...must be observed in at least two samples to activate allele frequency variation filtering”. This can also be the result of batch level contamination as samples are processed in 96 well plates and multiplexed. An additional feature I think your method brings is that it could be used as a QC check to detect contamination. Amplicon sequencing is extremely prone to contamination.

We in general agree with this point and one can further adjust this requirement of two or more samples. Nevertheless, it could happen that contamination within the lab is similar and thus would require further adjustment or pre-processing of the sequenced material.

R3Q6. In Figure 2B 'after applying variable', do the large green dots at the top of the plot fall within the primer regions (https://github.com/artic-network/artic-ncov2019/tree/master/primer_schemes/nCoV-2019/V3)? If so, you should treat them with extreme caution.

This feedback is related to the reviewer’s R3Q4. We appreciate the reviewer for pointing out that primer regions would introduce artifacts during the variant calling process. To address this issue, we have masked the primer regions for all COVID dataset that are using ARTIC v3 primers, and updated the results and figures in the manuscript.

For the details of the updated result, please refer to the response to R3Q4.

R3Q7. SARS-CoV-2 is the virus, COVID-19 is the disease, so you should correct the terminology.

We appreciate the detailed feedback. The terminology in the manuscript has been corrected.

R3Q8. Please note the reference genome you used (Wuhan Hu-1 ?).

Indeed. All COVID sequences are aligned to Wuhan Hu-1 strain of SARS-CoV-2. We have updated the manuscript to include the information on the reference genomes for all datasets in this study:

> (Methods, Quality control and read alignment, Page 14)

For time series dataset and cross-patient dataset for COVID samples, the reference genome used during the alignment is NCBI Reference Sequence: NC_045512.2. For the norovirus dataset, the reads were aligned to NCBI Reference Sequence: NC_039477.1. For the Ebola dataset, the reads were aligned to a reference strain from early in the associated outbreak (GenBank: KR817198.1). For the synthetic data, the reads were aligned to exclusively S and N gene regions of the reference genome (NCBI Reference Sequence: NC_045512.2).

Reviewer comments, second round

Reviewer #1 (Remarks to the Author):

The authors have generally done an excellent job of addressing the points raised by all three reviewers. They have gone to significant effort and their responses are thorough. I think the article benefits from the inclusion of the ebola & novovirus datasets, which broaden the potential utility of Variabel, and the analysis of synthetic controls. I am therefore happy to support publication of the paper once a final issue is addressed:

I previously suggested that the authors should implement their proposed large-scale database/framework from public SARS-CoV-2 data, as the availability of such a resource would greatly improve the utility of Variabel for SARS-CoV-2 genomics. In their response, the authors have explained very convincingly why this is not feasible and I tend to agree. However, I don't think it is fair to propose this solution in the manuscript without also identifying the numerous major hurdles that the authors have provided in their response, which will most likely ensure that this solution is never implemented.

This is the main section that I'm referring to:

"Based on a simple simulation (see Figure 4B) we calculate that approximately 10,000 samples would be required to recover most of the intra-host variants if we assume variants occur randomly along the genome of SARS-CoV-2. Similarly, we also expect a small drop in performance of Variabel if the time series data included fewer samples (e.g., 2-5). Both scenarios could be improved by leveraging a centralized data depository of low frequency SNV for SARS-Cov-2. Follow-up studies can then leverage this resource to assess and evaluate the biological importance of observed low frequency variants within and across hosts over time. However, established COVID databases such as GISAID are limited only to consensus level sequences 30,31, which might be a limiting factor going forward in this or future pandemic or outbreaks."

In my opinion, the authors should: (1) either remove the suggestion that a centralised repository could solve this issue for Variabel or very clearly explain why they have not implemented this solution themselves and; (2) make it very clear to the reader that, without this large and complex resource, the detection of low-AF variants in small-scale cross-patient SARS-CoV-2 datasets will not be very accurate.

I'm focusing on a negative here but overall I think this is an interesting, well-written paper and I congratulate the authors on the work. I hope to see this last point addressed and the article promptly published.

Cheers
Ira Deveson

Reviewer #2 (Remarks to the Author):

The authors have made a number of very useful additions to the revised manuscript including expanding the breadth of their analysis to include ebola and norovirus datasets. While the underlying results appear robust, the choice of reference sequence for the norovirus dataset is not ideal. The prototype NCBI strain is genogroup I (Norwalk strain), while the ONT dataset is derived from genogroup II sequences. The level of divergence between GI & GII viruses could be between 40-60% (amino acid). My concern is not necessarily the ability for Variabel to identify sub-consensus variants but rather poor or incorrect alignments where GII derived reads have been mapped to the GI reference genome.

Could the authors please provide some assurance that the results haven't been impacted by the choice of this divergent reference sequence?

Reviewer #3 (Remarks to the Author):

The authors have addressed all of my comments satisfactorily. This is a great paper and I support its publication.

LEGEND

Responses in bold **black**

Changes to the manuscript in **blue**

Location of inserted text in (parenthesis)

Reviewer #1:

The authors have generally done an excellent job of addressing the points raised by all three reviewers. They have gone to significant effort and their responses are thorough. I think the article benefits from the inclusion of the ebola & novovirus datasets, which broaden the potential utility of Variabel, and the analysis of synthetic controls. I am therefore happy to support publication of the paper once a final issue is addressed:

R1Q1: I previously suggested that the authors should implement their proposed large-scale database/framework from public SARS-CoV-2 data, as the availability of such a resource would greatly improve the utility of Variabel for SARS-CoV-2 genomics. In their response, the authors have explained very convincingly why this is not feasible and I tend to agree. However, I don't think it is fair to propose this solution in the manuscript without also identifying the numerous major hurdles that the authors have provided in their response, which will most likely ensure that this solution is never implemented.

This is the main section that I'm referring to:

"Based on a simple simulation (see Figure 4B) we calculate that approximately 10,000 samples would be required to recover most of the intra-host variants if we assume variants occur randomly along the genome of SARS-CoV-2. Similarly, we also expect a small drop in performance of Variabel if the time series data included fewer samples (e.g., 2-5). Both scenarios could be improved by leveraging a centralized data depository of low frequency SNV for SARS-Cov-2. Follow-up studies can then leverage this resource to assess and evaluate the biological importance of observed low frequency variants within and across hosts over time. However, established COVID databases such as GISAID are limited only to consensus level sequences 30,31, which might be a limiting factor going forward in this or future pandemic or outbreaks."

In my opinion, the authors should: (1) either remove the suggestion that a centralized repository could solve this issue for Variabel or very clearly explain why they have not implemented this solution themselves and; (2) make it very clear to the reader that, without this large and complex resource, the detection of low-AF variants in small-scale cross-patient SARS-CoV-2 datasets will not be very accurate.

We appreciate the reviewer's valuable feedback. We have added the following text in the manuscript to emphasize that the detection for low allele frequency variants for small

scale cross-patient dataset will be limited without a centralized repository for low frequency SNVs for SARS-Cov-2.

> (Main Text, Page 6)

Given established viral genomic databases such as GISAID are limited only to consensus level sequences, coordinated community efforts to store and track low frequency variation across vast collections of SARS-CoV-2 datasets will significantly boost Variabel's ability to detect low-AF variants in cross patient samples.

Reviewer #2:

The authors have made a number of very useful additions to the revised manuscript including expanding the breadth of their analysis to include ebola and norovirus datasets.

R2Q1: While the underlying results appear robust, the choice of reference sequence for the norovirus dataset is not ideal. The prototype NCBI strain is genogroup I (Norwalk strain), while the ONT dataset is derived from genogroup II sequences. The level of divergence between GI & GII viruses could be between 40-60% (amino acid). My concern is not necessarily the ability for Variabel to identify sub-consensus variants but rather poor or incorrect alignments where GII derived reads have been mapped to the GI reference genome.

Could the authors please provide some assurance that the results haven't been impacted by the choice of this divergent reference sequence?

We thank the reviewer for the feedback. With respect to the reviewer's concern about choosing correct reference genome, the norovirus dataset is a cross-patient dataset with norovirus GII-positive samples. The reference genome we used for read alignment of this dataset is NCBI Reference Sequence: NC_039477.1 (Norovirus GII, complete genome), which we stated in the manuscript (Page 14, Methods section, Quality control and read alignment subsection)

Although the results have not been impacted since the norovirus reference genome used for read alignment matched the norovirus in the samples, given this generated confusion, we have updated the following text in the manuscript for clarification.

> (Methods, Dataset descriptions, Page 13)

The norovirus dataset includes 39 full-length amplicon sequenced cross-patient norovirus GII-positive samples on ONT sequencing platform.

> (Methods, Quality control and read alignment, Page 14)

For the norovirus dataset, the reads were aligned to the norovirus GII reference genome (NCBI Reference Sequence: NC_039477.1)

Other changes to the manuscript

We changed multiple “cross patient” to “cross-patient” in the manuscript text, figure titles, and figure legends for consistency.

We added multiple “the” to the description of dataset in the manuscript text, figure titles, and figure legends for consistency.

We have added some acknowledgements to the manuscript.

> (Acknowledgements, Page 17)

We would like to thank Dr. Michael Nute for his constructive feedback on the project.

> (Acknowledgements, Page 17)

This work was supported in part by the Big-Data Private-Cloud Research Cyberinfrastructure MRI-award funded by NSF under grant CNS-1338099 and by Rice University's Center for Research Computing (CRC).